# Analyses of Genes Critical to Tumor Survival Reveal Potential ‘Supertargets’: Focus on Transcription

**DOI:** 10.3390/cancers15113042

**Published:** 2023-06-03

**Authors:** Darya Chetverina, Nadezhda E. Vorobyeva, Balazs Gyorffy, Alexander A. Shtil, Maksim Erokhin

**Affiliations:** 1Group of Epigenetics, Institute of Gene Biology, Russian Academy of Sciences, 34/5 Vavilov Street, Moscow 119334, Russia; chetverina@genebiology.ru; 2Group of Dynamics of Transcriptional Complexes, Institute of Gene Biology, Russian Academy of Sciences, 34/5 Vavilov Street, Moscow 119334, Russia; 3Departments of Bioinformatics and Pediatrics, Semmelweis University, H-1094 Budapest, Hungary; 4Cancer Biomarker Research Group, Research Centre for Natural Sciences, Institute of Enzymology, H-1117 Budapest, Hungary; 5Blokhin National Medical Research Center of Oncology, 24 Kashirskoye Shosse, Moscow 115522, Russia; shtil@scamt-itmo.ru; 6Group of Chromatin Biology, Institute of Gene Biology, Russian Academy of Sciences, 34/5 Vavilov Street, Moscow 119334, Russia

**Keywords:** cancer, tumor markers, DepMap, DOCK5, BEST3, TEAD3, WDR88, NFIA, ZBTB18

## Abstract

**Simple Summary:**

A variety of anticancer therapeutic targets have been identified over the decades. Nevertheless, the complexity of biological regulation dictates the necessity of knowledge about mechanisms specific to a particular tumor type. Using the DepMap CRISPR/Cas9 knockout database, we performed a comprehensive search for genes critical for tumor survival. Both established and novel markers of tumor viability were identified, many of which are transcriptional regulators. Our results substantiate new therapeutic strategies applicable to individual tumors.

**Abstract:**

The identification of mechanisms that underlie the biology of individual tumors is aimed at the development of personalized treatment strategies. Herein, we performed a comprehensive search of genes (termed Supertargets) vital for tumors of particular tissue origin. In so doing, we used the DepMap database portal that encompasses a broad panel of cell lines with individual genes knocked out by CRISPR/Cas9 technology. For each of the 27 tumor types, we revealed the top five genes whose deletion was lethal in the particular case, indicating both known and unknown Supertargets. Most importantly, the majority of Supertargets (41%) were represented by DNA-binding transcription factors. RNAseq data analysis demonstrated that a subset of Supertargets was deregulated in clinical tumor samples but not in the respective non-malignant tissues. These results point to transcriptional mechanisms as key regulators of cell survival in specific tumors. Targeted inactivation of these factors emerges as a straightforward approach to optimize therapeutic regimens.

## 1. Introduction

Genetic factors of tumor progression are distinct for individual cancer types. In each case, there is a unique set of regulatory circuits (i.e., signaling pathways and transcriptional regulators) whose dysfunction contributes to the biology of specific malignancies. Commonly, conventional chemotherapeutics are indiscriminately toxic to many cell types including non-malignant counterparts. Ideally, for each tumor type, it is necessary to select the unique targets whose inactivation would preferentially affect the particular tumor [1,2].

A large-scale analysis of transformed cell lines has been demonstrated to be promising for the identification of genes essential for tumor cell proliferation/survival [3,4,5,6,7,8]. The most sizeable resource that accumulates high throughput data is the DepMap (Dependency Map) project (https://depmap.org/portal/, accessed on 15–30 April 2022 and on 17–19 April 2023). This database contains a comprehensive set of genes individually knocked out by CRISPR/Cas9 technology in a broad panel of human-transformed cell lines. Results are represented as the ‘gene effect’ score indicating the probability of dependency of each cell line on the gene of interest. Strong negative values mark the cases where a given gene is critically important for cell proliferation/viability [3].

In this study, we used the DepMap panel to search for genes whose knockout most specifically affected the viability of the particular tumor type in comparison with other tumors. We coined the term ‘Supertargets’ for these critical genes. In total, we analyzed cell lines derived from 27 tumor types (six hematological and 21 solid malignancies). The top five genes within each tumor type were identified on the basis of efficacy and selectivity for cell viability inhibition. Next, using the TNMplot database portal, we analyzed the expression of identified genes and uncovered the transcripts deregulated in tumor vs. normal samples. The majority of identified Supertargets have been previously implicated in tumor biology and/or served as therapeutic targets, thereby validating the adequacy of the approach. Importantly, our cohort of Supertargets also contains poorly studied genes with unknown roles. These genes deserve future in-depth investigation as potential new targets for anticancer therapy.

## 2. Materials and Methods

The DepMap website (https://depmap.org/portal/, accessed on 15–30 April 2022 and on 17–19 April 2023) analysis of the dependency of tumor cell lines on individual genes was conducted using the CRISPR (release CRISPR (DepMap 21Q3 Public+Score, Chronos; accessed on 15–30 April 2022) and RNAi (release RNAi (Achilles+DRIVE+Marcotte, DEMETER2); accessed on 17–19 April 2023). Gene effect difference was calculated as T-statistic scores by DepMap portal using scipy.stats.ttest_ind (https://docs.scipy.org/doc/scipy/reference/generated/scipy.stats.ttest_ind.htm, accessed on 15–30 April 2022). Gene expression analysis in cell lines was carried out by DepMap Expression Public 22Q4 release, accessed on 10–13 April 2023.

Gene expression in tumor samples and the respective normal tissues were evaluated with Mann–Whitney test using the TNMplot database (https://tnmplot.com/, accessed on 20–25 December 2022), which contains transcriptome data from The Cancer Genome Atlas (TCGA) and The Genotype-Tissue Expression (GTEx) repositories [9]. GO enrichment analysis was performed using The Gene Ontology Resource (http://geneontology.org/, accessed on 20 January 2023). Annotation version and release date: GO Ontology database DOI: 10.5281/zenodo.6799722 (released 1 July 2022). Analysis type: PANTHER Overrepresentation Test (released 13 October 2022); test type: Fisher’s exact [10]. The following annotation datasets were used: ‘GO cellular component complete’ to describe the localization of Supertargets in cells; ‘GO PANTHER pathways’ to characterize signaling pathways; and ‘PANTHER GO-Slim Molecular Function’ to identify functional classes of Supertarget gene products.

The GOplot analysis was carried out using the TNMplot database (https://tnmplot.com/, accessed on 15 April 2022). DNA-binding domains of transcription factors were determined using Homo sapiens Comprehensive Model Collection (HOCOMOCO, https://hocomoco11.autosome.org/, accessed on 15 January 2023 [11]). The search for references for each gene was carried out using PubMed (https://pubmed.ncbi.nlm.nih.gov, accessed on 20–25 May 2022). The survival analysis was carried out using the Pan-Cancer datasets of the online tool www.kmplot.com (accessed on 20 April 2023).

## 3. Results

### 3.1. Approach

In the DepMap project, a cell line is considered dependent if the probability of dependency is <0.5; a score of 0 is equivalent to a non-essential gene whereas a score of −1 corresponds to the median of all commonly essential genes. Appendix A shows three major situations depending on the molecular function of the given gene. Disruption of the *RPL3* gene encoding the ribosomal L3 protein is lethal in all tested cancer lines. As expected, deletion of the *HTR1B* gene coding for the neurotransmitter receptor of the release of serotonin, dopamine, and acetylcholine in the brain, had minimal cytotoxic effects in tumor cells. At the same time, disruption of the *MYB*, a well-studied factor in blood malignancies (see below), is insufficient for a majority of cancer cell lines, although *MYB* inactivation has a strong inhibitory effect in hematopoietic tumors.

The difference in average ‘gene effect’ values between the selected group and the rest of the cell lines can be counted as a T-statistic score (see Material and Methods). The lowest T-statistic scores correspond to genes whose knockout affects cell viability in the particular tumor type. We selected the top five genes (referred to as Supertargets) with the lowest T-statistic scores for 27 cancer types using the DepMap portal. Since T-statistics scores depend on the number of cell lines for the respective tumor type, in each cohort we analyzed the top five genes independently.

### 3.2. Blood Malignancies

#### 3.2.1. Supertargets in Acute Myeloid Leukemia (AML) and Acute Lymphocytic Leukemia (ALL)

To further validate the proposed approach, we focused on AML and ALL, the malignancies with well-characterized therapeutic targets [12,13]. Indeed, the top five targets for AML and four out of the top five ALL targets have been described (Figure 1A). Details of Supertargets for each tumor type are given in Appendix A.

The *MYB* gene encoding the MYB transcription factor is deregulated in hematological malignancies including AML [14] and ALL [15,16]. In our analysis, this marker was identified with a minimal T-statistic score (rank 1) for AML and the second minimal score (rank 2) for ALL cells. The MYB protein binds to DNA via the helix-turn-helix (HTH) type domain and acts in cooperation with the CBP/p300 co-activator complex essential for the maintenance of the proliferative potential in AML cells [17]. In the mouse AML model, shRNA-mediated MYB down-regulation resulted in the remission of leukemia without inhibition of normal myelopoiesis [18]. Several chemical compounds have been developed to preferentially inhibit the proliferation of AML vs. normal hematopoietic cells (see [19] for a detailed review).

Other AML Supertargets encoded SPI1 (SPI1 proto-oncogene, a.k.a. PU.1), LIM domain only 2 (LMO2), Janus kinase 2 (JAK2), and growth factor independence 1 (GFI1) proteins (Figure 1A). The PU.1 transcription factor belongs to the erythroblast transformation-specific (ETS) family important for hematopoiesis. The role of PU.1 in AML is the subject of research [20]; PU.1 elimination was therapeutically beneficial in leukemia [21]. LMO2 is a key hematopoietic regulator that functions as a bridging molecule in the multiprotein transcription activator complex that includes, in addition to LMO2, the AL1/SCL, GATA1 and LDB1 subunits [22,23]. Chromosomal translocations t(11;14) (p13;q11) and t(7;11) (q35;p13) activated *LMO2* in acute T-cell leukemia; to date, LMO2 function has been characterized mostly in this tumor type [24]. However, the DepMap data suggest that AML cells, rather than ALL counterparts, are much more sensitive to *LMO2* knockout (Appendix A). Furthermore, JAK2 is a member of the Janus family of non-receptor tyrosine kinases involved in the control of cell growth, proliferation, and differentiation [25]. The role of JAK2 in hematopoiesis and AML is well studied, and a number of JAK2-targeted small molecule inhibitors have been shown to be efficient in adult and pediatric AML [26]. Finally, GFI1 is a C2H2 type zinc finger transcriptional repressor implicated in the pathogenesis of AML and myelodysplastic syndrome (MDS). Presumably, GFI1 might be involved in the progression from MDS to AML [27].

The *LEF1* (lymphoid enhancer binding factor 1), *RUNX1* (RUNX family transcription factor 1), and *EBF1* (EBF transcription factor 1) genes, identified as ALL Supertargets, are key hematopoietic transcription regulators that play central roles in differentiation and survival of lymphocyte progenitors [28] (Figure 1A). Apart from the above factors, the gene with minimal T-statistic score in ALL is *ATP6V0A2* (Figure 1A). To our knowledge, this marker is unknown in ALL biology and is therefore a new candidate target. The *ATP6V0A2* gene encodes the component of the proton channel of vacuolar ATPase (V-ATPase). A recent study indicated that the V-ATPase complex, in addition to its main function in generating the electrochemical proton gradient across the membranes, is involved in Notch/Wnt-dependent tumor progression [29].

Next, we investigated whether the transcripts of identified genes were differentially presented in tumor vs. normal tissues. Among the hematological malignancies, the expression data for AML and ALL are available at the TNMplot portal [9]. Figure 1B shows the comparison of the Supertarget gene expression between normal and tumor clinical samples. For nine out of ten genes, the expression in tumor samples was significantly higher than in the respective normal counterparts. The highest increase was detected for *MYB* mRNA in AML (FC median = 153.75) and ALL (FC median = 113.25). The expression of other genes also increased (FC ranged from 1.45 to 8.59). One exception was the *SPI1* mRNA decrease in AML (FC = 0.19; Figure 1B). The decrease in *SPI1* transcripts in AML samples compared to normal bone marrow cells is in accordance with the findings of a previous study [30]. Importantly, even when AML cells have low amounts of *SPI1* transcripts, their growth was sensitive to *SPI1* inhibition by RNAi or chemical compounds [21]. To estimate gene effect differences between the transformed and non-malignant cells of the same tissue origin, we took advantage of the new 22Q4 DepMap release that accommodates data on non-transformed cells. In support of the importance of identified Supertargets, the knockout of the *MYB*, *SPI1*, *LMO2*, or *JAK2* genes was cytotoxic in AML lines but not in the respective non-malignant cells (Appendix A). Thus, the DepMap ‘gene effect’ criterion is adequate for the accurate identification of known factors important for the proliferation/survival of AML and ALL cells, suggesting that this approach is straightforward for the search of Supertargets in hematological malignancies.

#### 3.2.2. Chronic Myelogenous Leukemia (CML), Lymphoma, and Multiple Myeloma

All CML cell lines in the DepMap panel contain the Philadelphia chromosome that leads to the formation of BCR-ABL1 fusion proteins. As expected, at the top of CML Supertargets are the *BCR* and *ABL1* genes with extremely low T-statistic scores (Figure 2A). Furthermore, three CML Supertarget genes, that is, GRB2-associated binding protein 2 (*GAB2*), son of sevenless homolog 1 (*SOS1*), and signal transducer and activator of transcription 5B (*STAT5B*), are the components of JAK/STAT signaling pathway critical in BCR-ABL1-positive neoplasms [31,32,33,34].

The most important genes in Hodgkin’s lymphoma (HL; Figure 2B) encode proteins implicated in signal transduction: interleukin-13 receptor subunit alpha-1 (IL13RA1) and its direct partners, STAT6 and IL4R (reviewed in [35]). Two genes with the lowest T-statistic scores encode the transcriptional repressors that regulate lymphogenesis: Ikaros zinc finger protein 1 (*IKZF1*) [36] and the basic leucine zipper transcription factor ATF-like 3 (*BATF3*) [36].

In non-Hodgkin’s lymphoma (NHL), four out of five Supertargets (myocyte enhancer factor 2B (*MEF2B*), EBF transcription factor 1 (*EBF1*), BCL6 transcriptional repressor (*BCL6*), and paired box 5 (*PAX5*)) encode well-studied transcription factors with significant roles in B-cell malignancies (reviewed in [37]) (Figure 3A). The marker ranked fourth is the *SH3GL1* gene that encodes the ubiquitously expressed endophilin-A2 implicated in endocytosis [38]. To our knowledge, no data point to a direct role of endophilin-A2 in tumor biology.

In multiple myeloma (MM), four Supertarget genes encode transcription factors such as interferon regulatory factor 4 (IRF4), PR/SET domain 1 (PRDM1), POU class 2 homeobox associating factor 1 (POU2AF1), and myocyte enhancer factor 2C (MEF2C, Figure 3B). Products of these genes are implicated in transcriptional regulation during germinal center formation and pathogenesis of B-cell malignancies [39]. The fourth gene is a homocysteine inducible endoplasmic reticulum protein with the ubiquitin-like domain 1 (*HERPUD1*), a component of the quality control system of ubiquitin-dependent degradation of misfolded proteins. HERPUD1 is implicated in ovarian [40] and liver [41] cancers, whereas no data link HERPUD1 to MM. Thus, we identified two new Supertargets, i.e., *SH3GL1* for NHL and *HERPUD1* for MM.

### 3.3. Supertargets in Solid Tumors

The search of Supertargets was performed for 21 solid tumor types presented in the DepMap panel. Figure 4A shows the T-statistic scores for the top five genes whose knockout was lethal. Of note, skin malignancies and neuroblastoma were the most sensitive, whereas the liver and osteosarcoma cells were less affected.

We next analyzed the differential expression in clinical solid tumor samples using the TNMplot portal. Figure 4B summarizes the RNAseq TNMplot data for clinical samples. In total, more than half of the identified Supertargets were significantly overexpressed in tumor samples (*p* < 0.01, Mann–Whitney test). The most dramatically elevated transcripts of the top five genes were observed in neuroblastoma (FC ranging from 218 to 4085). Additionally, in breast cancer and osteosarcoma, all the top five Supertargets were up-regulated. In contrast, *SMARCA2* (a subunit of the SWI/SNF chromatin remodeling complex) and *DDX5* (RNA helicase) (Figure 4B) were significantly down-regulated in lung cancer samples. Interestingly, while the ovarian and renal cell carcinoma (RCC) cells were sensitive to *PAX8* knockout, the transcription profiles of *PAX8* in normal and tumor samples differed for each of these malignancies (Figure 4B). In the normal ovary, *PAX8* transcription was below the level of detection, although drastically increased in tumors (Appendix A). The *PAX8* transcripts were elevated in normal kidneys but were down-regulated in RCC (Appendix A).

We next focused on the tumor types in which the expression of all Supertargets was significantly increased in clinical samples: neuroblastoma, breast cancer, and osteosarcoma (Figure 4B, Appendix A). In neuroblastoma, our analysis identified the transcription factors ISL1, HAND2, PHOX2B, PHOX2A, and MYCN as the top five targets critical for the survival of cell lines (Appendix A). *MYCN* is a molecular hallmark of this tumor, thereby justifying the significance of the other four markers. Indeed, MYCN, ISL1, HAND2, and PHOX2B, together with GATA3 and TBX2, comprise the core regulatory circuit, an autoregulatory transcriptional loop that maintains the malignant phenotype of MYCN-positive neuroblastoma [42]. One may hypothesize that the increased MYCN abundance is a result of gene amplification or overexpression as an early trigger. At physiological levels, MYCN binds to gene promoters; however, if abundant, this transcription factor can invade enhancers. The HAND2 partner cooperates with MYCN to compete with nucleosomes in the regulation of global gene transcription. These events are sensitive to down-regulation of their upstream mechanisms: a combination of Aurora kinase A and histone deacetylase inhibitors suppressed the tumor [43]. In line with these observations, Li et al. demonstrated that inhibition of Aurora kinase A led to ISL1 attenuation [44]. Thus, pharmacological combinations of selective protein kinase blockers with chromatin-targeting agents raise novel therapeutic possibilities.

Breast cancer was also the disease with substantially increased transcription of each of the five Supertargets (Figure 4B, Appendix A). *SPDEF* (ranked 1) contains the ETS DNA binding domain, a subtype of the bHLH domain; functions of this gene product are vital for normal development as well as for the survival of breast cancer cells [45]. The Supertarget genes, *FOXA1*, *ESR1*, and *GATA3,* are among the most studied prognostic markers and therapeutic targets in breast cancer (reviewed in [46,47]). As for the *TRPS1* (transcriptional repressor GATA binding 1) gene, its overexpression has been recently shown to drive genome evolution in breast carcinomas [48].

In osteosarcoma, *SMARCAL1*, *IRS1*, *SUB1*, *HMGA2*, and *FANCM* were identified as Supertargets (Figure 4B, Appendix A). Three genes encode the proteins implicated in DNA repair. The ATP-dependent chromatin remodeling protein SMARCAL1 interacts with damaged replication forks to promote their stability [49,50]. FANCM acts as an anchor for DNA repair complex [51]; pharmacological inhibition of FANCM was selectively toxic for cancer cells that use alternative lengthening of telomeres [52]. SUB1 recognizes G-quadruplexes and DNA lesions, thereby activating double-strand break repair by stimulating the joining of non-complementary DNA ends and promoting genome stability [53,54]. IRS1 is a cytosolic adaptor protein involved in insulin receptor and insulin-like growth factor I receptor signaling. This protein can also act as a transcription regulator to support tumor growth by an incompletely defined mechanism [55]. HMGA2 is a transcription factor with the high mobility group DNA binding domain. HMGA2 mediates multiple mechanisms of tumor progression including stimulation of proliferation and inhibition of apoptosis (reviewed in [56]).

### 3.4. Expression in Tumor Cell Lines and Functional Types of Supertargets

Differential sensitivity of tumor cells to individual gene depletion can result from cell type-specific gene expression. Alternatively, if the gene is expressed ubiquitously, its product can be critically involved in cell type-specific metabolism. To validate these options, we analyzed the expression of 114 Supertargets using RNAseq DepMap. In total, 96.5% of Supertarget genes were expressed in >50% tumor cell lines using the log2(TPM + 1) threshold; 73.7% Supertarget genes were expressed in >50% tumor cell lines if we set a higher threshold log2(TPM + 2) to cut very low expressing genes (Appendix A). Thus, the expression of the majority of Supertargets is not confined to a particular tumor type.

However, the expression of certain Supertargets may vary between cell lines of different tissue origins. We estimated the relative expression of Supertargets by measuring the ratio between the median expression of Supertargets in cell lines of the specific tumor type (for which the Supertarget was identified) and its median expression in the total cohort of tumor cells. We found that 83% of Supertargets were expressed higher in the particular tumor type compared to their average median expression. Moreover, the expression of 35% Supertargets was at least 2-fold higher; strikingly, the amounts of *MYOG* and *MYOD1* transcripts were 50-fold higher. Thus, while transcription of Supertargets is not tumor type-specific, its relative expression is generally elevated.

GO cellular component analysis using the Gene Ontology Resource, http://geneontology.org/, accessed on 20 January 2023, revealed that 92 out of 124 (73%) unique Supertargets were present in the nucleus (fold enrichment = 1.96, raw *p*-value = 8.95 × 10^−16^, FDR = 1.66 × 10^−13^) and 58 (46%) were associated with chromosomes (fold enrichment = 5.12, raw *p*-value = 3.12 × 10^−27^, FDR = 3.18 × 10^−24^). Supertargets were involved in different signaling pathways; the principal enrichment of observed vs. expected values was revealed for JAK/STAT (raw *p*-value = 2.32 × 10^−4^, FDR = 3.72 × 10^−3^), interleukin (raw *p*-value = 8.52 × 10^−9^, FDR = 4.54 × 10^−7^), insulin/IGF pathway mitogen-activated protein kinase kinase/MAP kinase cascade (raw *p*-value = 1.35 × 10^−3^, FDR = 1.35 × 10^−2^), and p53 pathway feedback loops 2 (raw *p*-value = 3.50 × 10^−4^, FDR = 4.31× 10^−3^) (Appendix A).

The analysis of molecular functions by the GOplot tool indicated an overwhelming enrichment of transcription factors among Supertargets (Figure 5A). According to PANTHER GO-Slim Molecular Function analysis, 52 (41.3%) Supertargets possessed sequence-specific double-strand DNA binding activity (fold enrichment = 7.61, raw *p*-value = 6.36 × 10^−32^, FDR = 6.94 × 10^−30^; Appendix A). Importantly, transcription factors represent the largest group both in solid and hematopoietic cancers (Appendix A).

In accordance with the predicted tumor specificity, the GOplot analysis of biological pathways revealed the ‘lymphocyte and T-cell differentiation’ and ‘regulation of hemopoiesis’ groups as the most enriched Supertargets of blood tumors, and the ‘cell fate commitment’ and ‘gland, muscle, and mesenchyme development’ groups for solid tumors (Appendix A).

To further estimate the representation of DNA-binding domains in Supertargets, we used the *Homo sapiens* Comprehensive Model Collection (HOCOMOCO, [11]) classification. This resource divides DNA binding factors into ten classes with 34 groups of specific DNA binding motifs. One-half (17/34) of specific motif groups were found among Supertargets (Appendix A); seven groups were presented most frequently (Figure 5B). Thus, transcription factors comprise the leading group within the entire set of Supertargets. However, we observed no prevalence of individual DNA binding motifs. While 18 proteins had the HTH domain, and 9 proteins belonged to C2H2-type zinc fingers, they corresponded to only 4.4% and 1.3% of proteins carrying these motifs, respectively. Thus, transcription factors with different types of DNA-binding domains are the most representative Supertargets.

### 3.5. Clinical Utility of Supertargets

We next investigated the correlation between the Supertarget gene expression and patient prognosis using the KMplot portal (https://kmplot.com, accessed on 15 February 2022). Out of 27 tumor types in DepMap, the KMplot database contains data on 12 types. We analyzed the overall survival (OS) of patients based on 60 (12 × 5) Supertargets. As a result, 17 genes demonstrated significant (*p* < 0.01) differences in life expectancy depending on the high or low expression of individual Supertargets (Figure 6A). In seven cases the elevated mRNA abundance correlated with poor prognosis, whereas overexpression of 10 genes was associated with a longer lifespan. Figure 6 shows correlations with the highest (*CCNE1*, uterine cancer; *KRAS*, pancreatic cancer) (Figure 6B) and the lowest (*MYB* in AML, *ZER1* in cervical cancer) (Figure 6C) hazard ratios (HR). Intriguingly, high or low HR values per se do not necessarily indicate the prognostic significance (compare HR data for KRAS vs. MYB), further substantiating the complexity of cellular roles of individual Supertargets in specific contexts.

### 3.6. Significance of Supertargets Is Supported by RNAi DepMap Data

Besides the CRISPR analysis, DepMap includes RNAi data. However, these parameters cannot be compared directly, since CRISPR and RNAi contain different sets of tumor lineages. Additionally, the knockdown efficiency by RNAi differs significantly for different genes. We estimated the effect of RNAi knockdown for top5 CRISPR identified Supertargets using data on five blood malignancies (AML, ALL, MM, Hodgkin’s, and Non-Hodgkin’s lymphomas) and for five solid tumors (skin cancer, neuroblastoma, rhabdomyosarcoma, Ewing sarcoma, and colorectal cancer). In the solid tumors, the five selected cancer types are those having the lowest T-statistic values by the CRISPR analysis (Figure 4). Overall, 50 genes were analyzed.

To take into account the RNAi efficiency, the genes were divided into two subgroups using the DepMap database criterion: 25 genes with high prediction accuracy and 25 with low prediction accuracy. Next, we determined whether each gene in each group can be considered a Supertarget by RNAi analysis for CRISPR-determined tumor type. For each gene, the statistical significance of the negative gene effect difference was assessed using the confidence threshold *p* < 0.0005 set by DepMap. If the gene fit this criterion, it was considered a Supertarget and was given an “RNAi KD” score of “1”; otherwise, the score was “0”.

Figure 7 shows that each of the 25 genes from the high prediction accuracy group fit the established criteria, while in the low prediction accuracy group, only 7/25 genes passed the confidence threshold *p* < 0.0005. Thus, in the case of efficient RNAi, the genes act as Supertargets upon knockdown.

Figure 8 shows the effects of RNAi on top3 Supertargets with the highest value of prediction accuracy in hematological and solid tumors.

### 3.7. Novel Supertargets

The Supertargets identified herein included the previously characterized markers as well as proteins whose function has not been attributed to the respective tumor. To analyze the latter group, we performed a PubMed search of manuscripts with a text combination of the gene and the respective tumor type. We found that 24 out of 135 Supertargets have not been linked to the respective tumors (Figure 9). Among these 24 genes, only three encoded the DNA binding transcription factors: NFIA, ZBTB18, and TEAD3. NFIA is required for the proliferation of cervical cells, ZBTB18 for rhabdomyosarcoma, and TEAD3 for the urinary tract cells (Figure 9, Appendix A). ZBTB18 has been identified as a tumor suppressor in glioblastoma [57] and colorectal cancer [58]. TEAD3 is a component of the Hippo pathway implicated in a number of diseases including cancer [59,60]. Functions of the *NFIA* gene are better understood in brain development [61], although recent findings link its role to tumor progression [62]. Regarding non-DNA binding factors, at least three are known to be directly involved in the regulation of transcription; SUB1 (required for osteosarcoma cell proliferation) and EDF1 (the upper aerodigestive tract) are transcriptional coactivators, whereas SPTY2D1 (in the cervix) is a histone chaperone.

## 4. Discussion

In the present study, we used the DepMap database to search for genes that can serve as therapeutic targets for individual tumor types. For each of the total 27 tumor types, we identified the top five genes whose CRISPR/Cas9 mediated knockout was lethal. A similar inhibitory effect was observed through the analysis of RNAi DepMap data. Importantly, we found 24 new Supertarget genes whose functions have not been investigated previously in the respective tumor.

Selectivity of Supertargets for an individual tumor can arise from two mechanisms. In the first scenario, the expression of Supertargets is cell type-specific. Second, the expression of Supertargets is ubiquitous, but its knockout affects cell type-specific cascades. Using RNAseq data available at the DepMap portal, we showed that, although transcription of Supertargets is not confined to a particular tumor type, the relative abundance of specific transcripts is higher in those cells in which the Supertarget gene was identified. Furthermore, using the TNMplot portal, we demonstrated that the expression of >50% of identified genes was significantly increased in matched tumor vs. non-malignant clinical samples. The most dramatic transcriptional burden was observed in neuroblastoma samples (a 200–4000-fold increase in steady-state levels of five Supertarget mRNAs).

Interestingly, genes coding for transcription factors with various DNA-binding domains are the predominant group within Supertargets (41%). Our data substantiate the key role of transcription factors in tumor biology [63,64]. For a long time, transcription factors have been considered non-druggable in contrast to enzymes or other biomolecules that bind small molecular weight compounds [65]. Recent advances in targeting transcription factors have changed this postulate and opened the way for new therapies using drugs against these proteins (reviewed in [66]). For example, the small molecule inhibitor AI-10-49 selectively disrupted the interaction between the aberrant activator CBF-β and DNA binding transcriptional factor RUNX1, thereby restoring normal expression patterns and delaying leukemia progression in mice [67]. Another example of a protein–protein interaction inhibitor is the small molecules that disrupt MDM2–p53 binding to reduce p53 ubiquitination, thereby increasing p53 abundance and cell death [68,69,70]. A series of PROTAC (PROteolysis Targeting Chimera)-based drugs targeting transcription factors have entered clinical trials [71,72,73].

From this viewpoint, three newly identified DNA binding Supertargets, NFIA, ZBTB18, and TEAD3, are of particular interest. They are required for the proliferation of cervical, rhabdomyosarcoma, and urinary tract tumors, respectively. In addition, several new Supertargets encoding non-DNA binding transcriptional regulators were identified for osteosarcoma (transcriptional coactivator SUB1), cervix (histone chaperone SPTY2D1), and upper aerodigestive tract (transcriptional coactivator EDF1). In additional, several other non-DNA binding transcriptional regulators are present among all Supertargets: a component of the cohesion complex STAG1 (Ewing sarcoma), SMARCA2 (lung) and SMARCAL1 (osteosarcoma). SMARCA2, a.k.a. BRM, is an ATPase subunit of SWI/SNF remodeling complexes [74,75], whereas SMARCAL1 is an SWI/SNF-related ATPase [50]. The presence of these proteins among Supertargets supports an important role for chromatin regulators and epigenetic markers in cancer progression [76]. The design of targeted chemical tools to selectively remove transcriptional regulators emerges as a challenging strategy to inactivate these intractable mechanisms.

Among the discovered Supertargets are the genes for which the oncogenic driver mutations have been found. Individual mutations can cause tumor hypersensitivity to deletion of the mutated gene; for example, GOF mutations of the EZH2 methyltransferase gene in lymphoid tumor cells [77]. A similar situation is observed in the case of KRAS, BRAF and CTNNB1 (Appendix A). Deletion of the KRAS gene is most dramatic for the survival of pancreatic cancer cells (Supertarget) as well as for lung and colorectal cancers (Appendix A). Forty-four KRAS mutations were detected in 47 pancreatic cell lines. The same situation was observed for the BRAF and CTNNB1 genes. The skin tumor lines with the mutant BRAF (Supertarget) were most sensitive to deletions of this gene. In addition, BRAF mutations were associated with the sensitivity of thyroid cancer cell lines (Appendix A). The CTNNB1 gene emerged as a factor of viability in colorectal cancer (Supertarget) and, to a lesser extent, in stomach cell lines (Appendix A). More studies are required to estimate in detail the relation of oncogenic driver mutations in Supertargets to tumor cell viability.

Intriguingly, the involvement of some Supertargets in tumor biology has been shown earlier, although for unrelated malignancies. For example, an aberrant *HERPUD1* expression has been reported for ovarian [40] and liver [41] cancers. However, the DepMap database demonstrated that *HERPUD1* gene knockout is most crucial in MM. Likewise, TRPM7 has been implicated in breast, pancreatic and gastric malignancies (reviewed in [78]). Nevertheless, our analysis suggests a major role of this protein in bile duct cancer. These results expand the role of individual markers beyond the specific tumor, thereby uncovering new mechanisms and suggesting previously neglected therapeutic opportunities.

Several new Supertargets remain poorly characterized; therefore, little is known about their functional properties. In glioma cells, this study identified *RPP25L* as a novel Supertarget. Currently, there is only one report that attributed the *RPP25L* gene product to RNase P/MRP complexes involved in tRNA processing [79]. In uterine carcinoma cells, one Supertarget was the *WDR88* gene encoding a protein with six WD40 repeat domains; however, information about its functions is lacking.

Finally, we provide initial evidence in support of the clinical relevance of Supertargets. Although the dataset for analysis is currently limited to 12 tumor types, a subgroup of Supertargets showed promising value in OS prognosis. However, it is premature to indiscriminately use Supertargets as prognostic markers because of data shortage and the complexities in interpretation. Indeed, the role of an individual Supertarget should be considered with regard to the specific context, largely the tissue origin of the tumor. One would expect that the patterns of gene expression significantly vary in the course of tumor development and the response to treatment. Therefore, the differential ranking and representation of individual Supertargets should be kept in mind as a prerequisite for evaluating their practical usefulness.

## 5. Conclusions

Our comprehensive search of critically vital genes, termed Supertargets, in the panel of tumors of various origins identified both established and yet unknown mechanisms. Importantly, the majority of proteins encoded by Supertargets appeared to be transcriptional regulators such as DNA binders and chromatin modifiers. This finding justifies the relevance of chemical tools that target transcription factors for treatment optimization. The development of selective instruments that combat the specific transcription regulators would help elucidate the functional roles of the identified genes. In turn, these tools would serve as prototypes of specific inhibitors for personalized therapies.

## Figures and Tables

**Figure 1 cancers-15-03042-f001:**
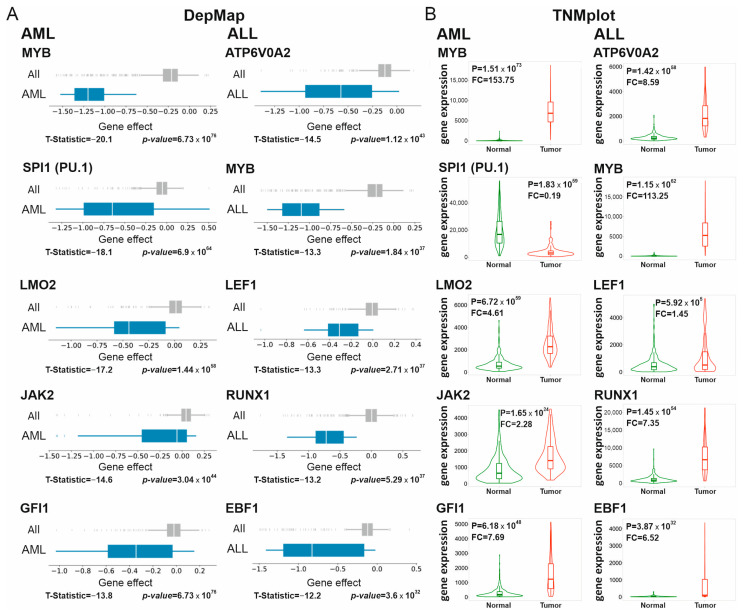
Analysis of gene effect difference reveals Supertarget genes in leukemia cell lines. (**A**) DepMap data analysis. Gene deletion effects for all cell lines are colored in grey (All, 1032 cell lines), and gene effects for tumor type-specific cell lines are rendered in blue. 26 and 16 cell lines were analyzed in case of Acute Myeloid Leukemia (AML) and Acute Lymphocytic Leukemia (ALL), respectively. The T-statistic scores and *p*-values are given below the diagrams. (**B**)The differential gene expression in clinical samples from TNMplot database. The differential gene expression in normal (green) vs. tumor (red) samples is presented as violin plots (P, *p*-value, Mann–Whitney test. FC, fold change median).

**Figure 2 cancers-15-03042-f002:**
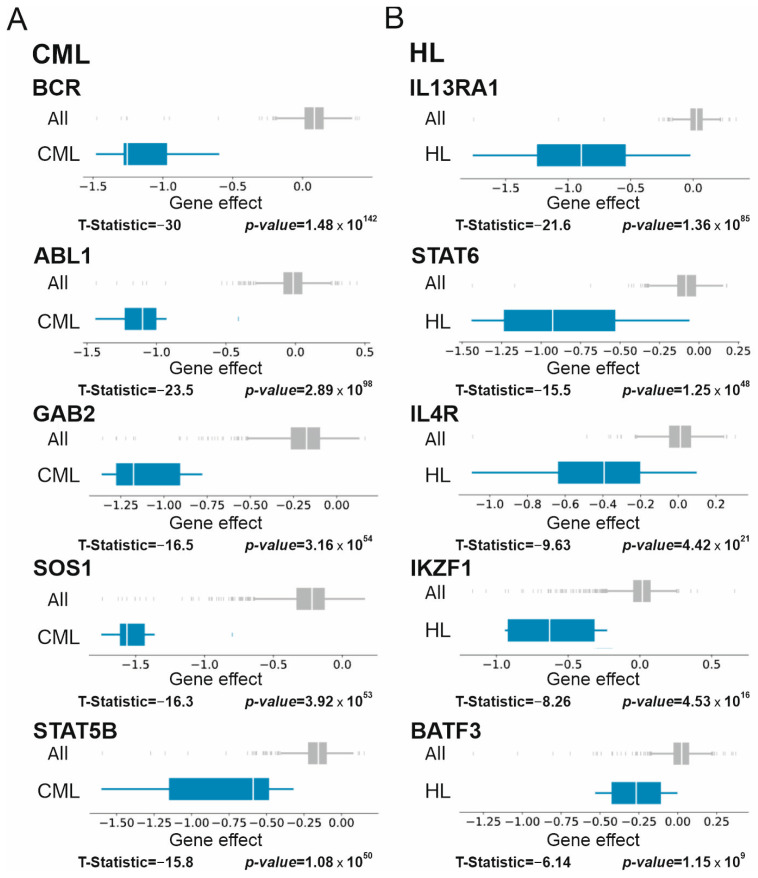
Supertarget genes in chronic myelogenous leukemia (CML, (**A**), 7 cell lines) and Hodgkin’s lymphoma (HL, (**B**), 4 cell lines) cell lines. Designations are as in Figure 1. See text for details.

**Figure 3 cancers-15-03042-f003:**
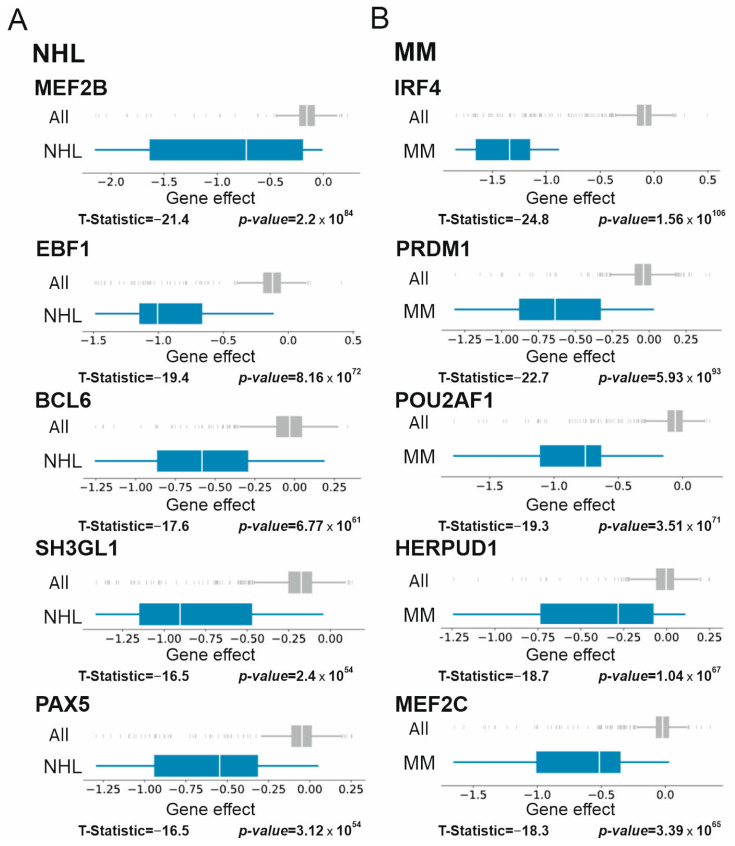
Supertarget genes in non-Hodgkin’s lymphoma (NHL, (**A**), 24 cell lines) and multiple myeloma (MM, (**B**), 21 cell lines) cell lines. Designations are as in Figure 1. See text for details.

**Figure 4 cancers-15-03042-f004:**
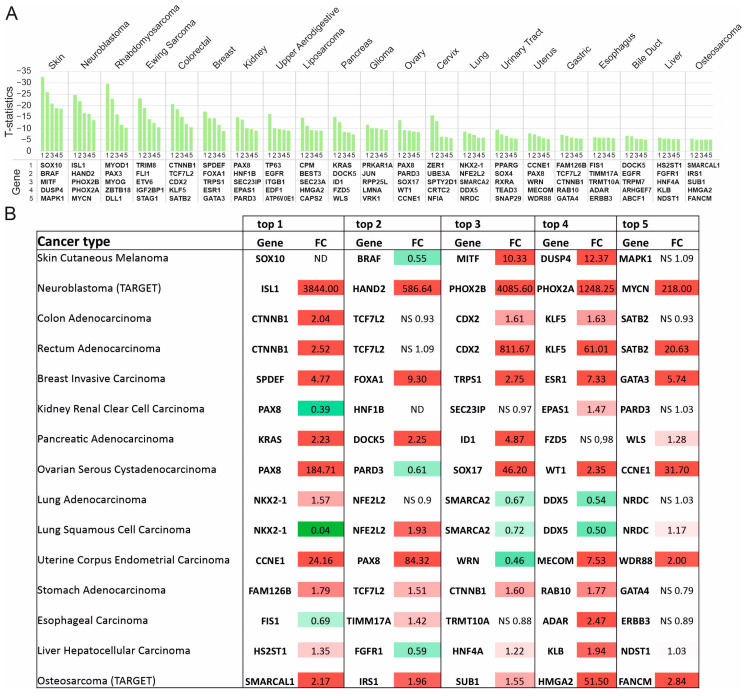
Supertargets in solid tumors. (**A**) Overview of gene effect difference calculated as T-statistic score for genes 1–5 in each cohort. (**B**) Differential gene expression (FC median; TNMplot database) in normal vs. tumor samples. Statistical significance was calculated by a Mann–Whitney U-test and set at 0.01. NS, non-significant; Mann–Whitney *p*-value. FC values for significantly overexpressed and down-regulated genes are highlighted in red or green, respectively. ND, no data. TARGET, pediatric tumors.

**Figure 5 cancers-15-03042-f005:**
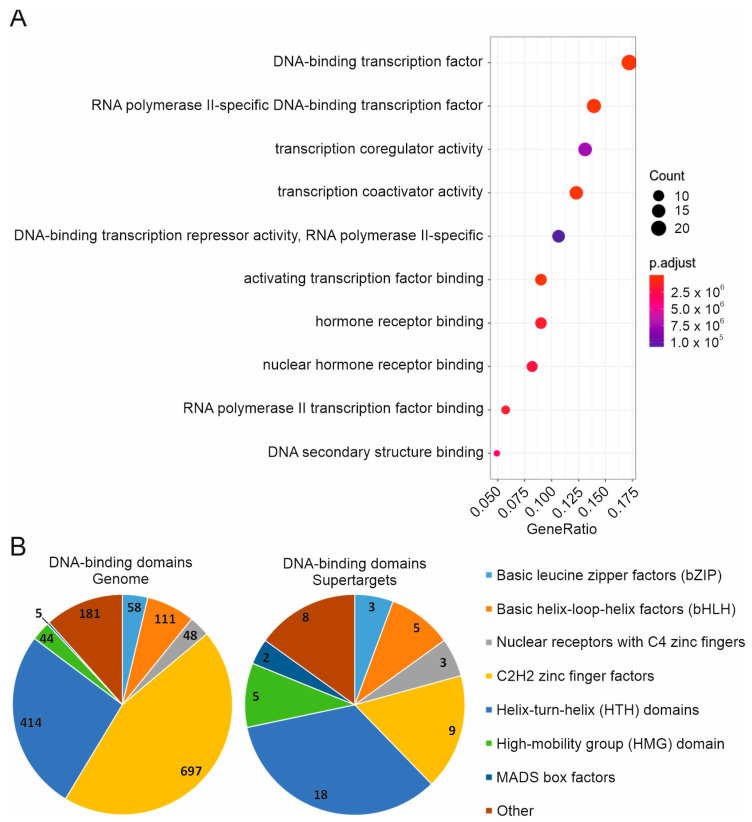
GO analysis of molecular functions of identified Supertargets. (**A**) GOplot molecular functions. The TNMplot resource was used (https://tnmplot.com/analysis/, accessed on 15 April 2022). (**B**) Types of DNA binding motifs across the genome (**left**) and in Supertargets (**right**).

**Figure 6 cancers-15-03042-f006:**
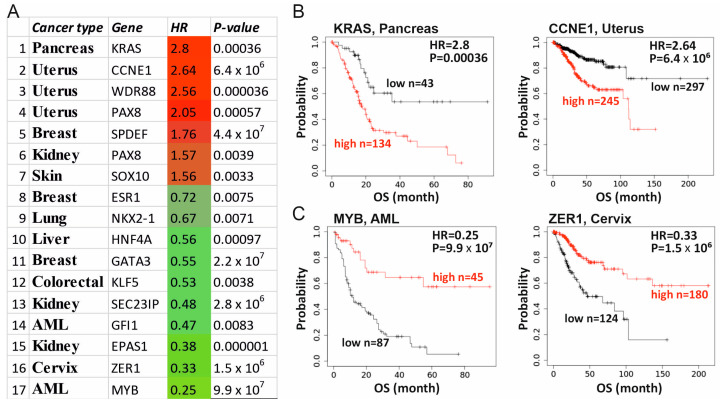
Prognostic value of Supertargets analyzed by KMplot resource. (**A**) Seventeen genes revealed significant (*p* < 0.01) differences in OS depending on high or low Supertarget expression. The hazard ratio (HR) is shown in red when higher gene expression is correlated with a shorter survival, and in green in case of a longer survival. (**B**) KMplots for *KRAS* (pancreatic cancer) and *CCNE1* (uterine cancer) genes with the highest HR. (**C**) KMplots for *MYB* (AML) and *ZER1* (cervix cancer) genes with the lowest HR. Red is the cohort with high gene expression; black, low expression.

**Figure 7 cancers-15-03042-f007:**
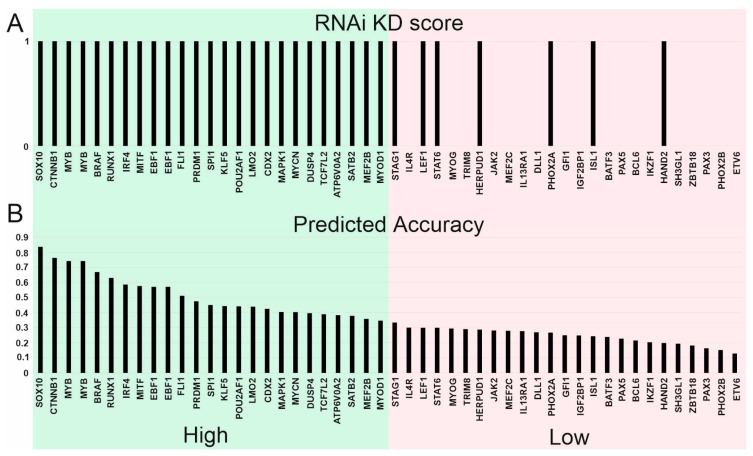
DepMap RNAi data analysis corroborates the efficacy of Supertargets identified by CRISPR knockout. (**A**) Conformity of Supertargets identified by CRISPR analysis with RNAi knockdown (KD) data. RNAi score 1: KD has an inhibitory effect in a particular tumor type (*p* < 0.0005); score 0: KD has no significant effect. (**B**) Prediction accuracy score for RNAi data. The high prediction accuracy group is highlighted in green; the low prediction accuracy is shown in pink.

**Figure 8 cancers-15-03042-f008:**
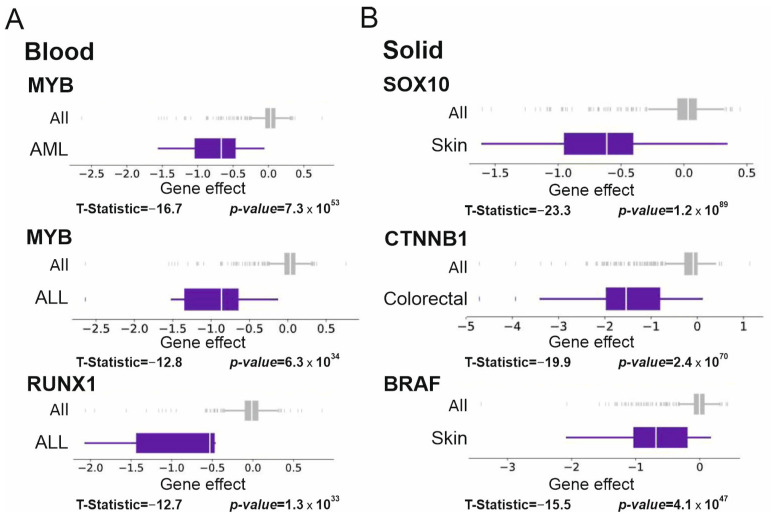
RNAi data (DepMap project) for blood and solid tumors. Gene depletion effects for all cell lines (All, 710 cell lines) are colored in grey; gene effects for tumor-specific cell lines are rendered in purple. T-statistic scores and *p*-values are given below the diagrams. (**A**) RNAi data for *MYB* gene in AML and ALL, and *RUNX1* gene in ALL tumor cell lines are shown. 22 and 7 cell lines were analyzed in case of AML and ALL, respectively. (**B**) RNAi data for *SOX10* gene in Skin, *CTNNB1* in Colorectal, *BRAF* in Skin tumor cell lines are shown. 47 and 43 cell lines were analyzed in case of Skin and Colorectal tumors, respectively.

**Figure 9 cancers-15-03042-f009:**
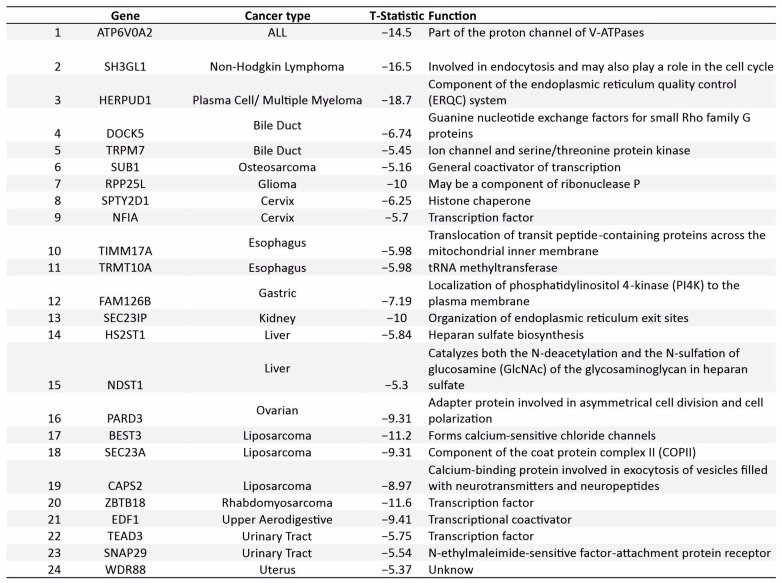
Newly identified Supertargets for blood and solid malignancies. Shown are Supertargets that have not been investigated in respective diseases.

## Data Availability

All relevant data are within the paper and Appendix A.

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
