# Peer review of "Analyses of Genes Critical to Tumor Survival Reveal Potential ‘Supertargets’: Focus on Transcription"

_cancers, 2023, doi:10.3390/cancers15113042_

Round 1

Reviewer 1 Report (Previous Reviewer 2)

The manuscript has been revised and can be accepted for publication.

Author Response

We thank the reviewer for this positive evaluation.

Reviewer 2 Report (New Reviewer)

The manuscript by Chetverina et al describes the identification of potential ‘supertargets’ for cancer therapy. The study is based on available Depmap CRISPR data and TNM (TCGA + GTEx) data as well as other WEB available portals. This pure bioinformatics-based manuscript described intelligently the use of available public resources to address an important question related to cancer research, namely identification of potential cancer type specific treatment targets.

Major points.

The main limitation of the manuscript is the lack of development of novel resources but rather demonstrate the specific use of existing public WEB-platforms for biomedical research. It shall be acknowledge that the latter is presented in form of interesting results. However, it is unclear how the authors have reached the conclusion that exactly the ‘top five genes’ in terms of the used T-statistic scores for each cell line is a reasonable cut-off. Why not top 10? or top 25? or a common T-statistic score cut-off for all tumor cell types?  The authors should elaborate on this and make it clear how the eventual extension of the number of called ‘supertargets’ for each tumor cell type would interfere with the overlap between ‘supertargets’ between tumor cell types.

It is not straightforward to understand if the ‘supertarget’ T-statistics scores depends on the number of cell lines for each cancer type (e.g. will a ‘supertarget’ for a tumor type represented with a high N of cell lines per se have a better T-statistics score than a ‘supertarget’ for a tumor type represented with a low N of cell lines). Please specify this and the implications.

Expected ‘supertargets’ would be genes harboring oncogenic driver mutations in cell lines for a given tumor type and already known to be potential therapy targets given the cells depends on these for growth/survival. This is to some extend demonstrate by the data for CML cell lines in figure 2A, all possessing BCR-ABL-fusions. However, it could be highly interesting for more proof-of-principle to demonstrate this in more mutation heterogeneous tumor cell populations. Could the authors make sub-grouping of cell lines for a specific cancer type (e.g. creating sub-groups of lung tumor cells with KRAS or EGFR mutations) and demonstrate that these oncogenes indeed (or not) are ‘supertargets’ herein. At least, it should be discussed why commonly mutated oncogenes (as for example the beforehand mentioned KRAS or EGFR oncogenes) only are sparsely identified as ‘supertargets’.

Minor points.

For figure 2, figure 3, and figure 8, gene expression data should be illustrated as it is in figure 1.

Please specify ‘ns’ and ’ND’ in legend to figure 4. In this figure it appears a little surprising that e.g a change in NKX2-1 expression of 0.04 in LUSQ can be significant whereas several other changes of larger magnitude appears insignificant. Finally the figure appears to have the wrong usage of symbols ‘,’ relative to ‘.’ (this also appearing in figure 6 and figure 9).

ok

Author Response

Below please find our point-by-point responses.

Sincerely yours,
Dr. Maksim Erokhin

Reviewer 3 Report (New Reviewer)

Chetverina et al. used the DepMap CRISPR/RNAi datasets and the TNM plot database to identify genes (called "Supertargets") that are essential for tumors of specific origin. Several of these genes are known to be key molecules for tumor growth, but they also found new candidate genes that contribute to the growth of tumor cells of specific origin, such as ATP6V0A2 in ALL, SH3GL1 in NHL, and HERPUD1 in multiple myeloma. They focused on DNA-binding transcription factors, which are enriched in Supertargets, and found that gene expression levels of several Supertarget genes were associated with patient prognosis. The paper is generally interesting, technically sound but they did not perform a validation cohort using other datasets, which is the main point to revise. And there seems to be too much explanation about each Supertrget gene (it looks like a review article).

Minor points

- (Fig 1A; Fig 2A, B; Fig 3A,B and Fig 4B) How many cell lines were used for analysis? It is better to indicate the number of cell lines as you show in Supplementary Figure S2.

- (Supplementary Figure S3) What does orange lines, grey dot lines, and orange circles indicate? Please add the explanations in the figure legend.

Author Response

Please find our point-by-point response in the file attached.

Sincerely yours,
Dr. Maksim Erokhin

Round 2

Reviewer 2 Report (New Reviewer)

The authors have well-addressed all the concerns.

Reviewer 3 Report (New Reviewer)

Thanks for putting more efforts into this study.

This manuscript is a resubmission of an earlier submission. The following is a list of the peer review reports and author responses from that submission.

Round 1

Reviewer 1 Report

This study utilized the DepMap portal database to evaluate the top five most crucial Supertarget genes for different cancer types based on CRISPR-identified Gene effect, and evaluated their roles and potential as therapeutic targets by literature review. Furthermore, TNMplot was used to analyze the relative expression of individual Supertargets in normal and tumor tissues, extending the analysis from hematological and lymphoid malignancies to a total of 27  cancer types and emphasizing the novelty of the defined Supertargets. However, the following concerns have lowered the reference value of the article and require further explanation from the author:

  1. The authors chose the top five genes with the lowest T statistic score as Supertargets. However, a lower T statistic score does not necessarily mean that the gene has a more significant effect, but only indicates a significant difference from the reference group. If the selection is based solely on this criterion, the top five genes with the lowest T statistic score may not be representative in different cancers. For example, in cancers where only a few genes have a T statistic score compared to those where many genes have a T statistic score, the Supertargets defined based on this criterion may have different reference values.
  2. If the authors defined the Supertargets as the top 10 genes, all data interpretations in the manuscript may have significant differences. Perhaps using a proportion-based definition of Supertargets would be less biased.
  3. The author's literature review on the Supertargets defined by different cancer types is a valuable contribution in terms of referencing and integrating previous studies. However, it does not necessarily indicate that these Supertargets have significant implications for disease progression in specific cancer types.
  4. Indeed, the BCR-ABL fusion gene has a historical significance in the development and treatment of hematologic malignancies, and the Supertarget analysis also shows that it has the lowest T statistic score in blood cancers. However, some of the identified Supertargets may be essential for the survival of the cells, such as IKZF1 and PAX5, whether they are normal or cancerous. By comparing the gene's Gene effect in all cell lines with that in a particular cancer cell line, the authors can only show that the gene is crucial for that cancer type, but it does not mean that the gene is not highly expressed in normal blood cells or that its function is unnecessary for cell survival.
  5. The authors did not provide a specific justification for setting the TPM threshold at >0.1 in at least 90% of cell lines to define universal expression of a gene. It is possible that they chose this threshold based on their previous experience or other studies in the field, but this is not explicitly stated in the article. Alternatively, perhaps a threshold of >1 in 90% of cell lines may be more appropriate depending on the gene and cell type under investigation. The choice of threshold can have a significant impact on the interpretation of universal or cancer-specific expression of super-target genes. Therefore, it would be beneficial for the authors to provide a more comprehensive explanation of their threshold selection and provide additional evidence to support their decision.
  6. The author's selection of the top signaling pathways for functional analysis is not clearly explained in terms of their association with the development of cancer caused by Supertargets. Moreover, the rationale for why PDGF signaling pathway and EGF receptor signaling pathway were not included in the analysis is not explained. A more comprehensive explanation of the selection criteria and the association between the identified signaling pathways and Supertargets in the context of cancer development would strengthen the study's conclusions. The author may consider using tools such as GOplot or clusterProfiler to present the super targets related to biological functions, which would help readers better understand the potential biological roles of these super targets.
  7. The Kaplan-Meier plot and log-rank test based on gene expression clustering have been widely used in literature to evaluate the association between specific genes and cancer prognosis. As the Supertargets have a significant impact on cancer cell survival, could the authors provide information on the expression clustering of Supertargets in different cancer types and whether they are significantly associated with clinical outcomes of patients? This would be of practical value in applying this strategy to the clinic.

Reviewer 2 Report

The present manuscript by Chetverina et al on identifying super targets based on CRISPER knockout database is based on an interesting concept, However, I have few concerns that needs to be addressed:

1. The material and method section is unclear, what criteria was used for database search and how were genes selected.

2. How did you actually define super targets; was it just on the basis of dependency score from DepMap website or any additional information was also taken into account.

3. There are multiple other factors while describing a gene as a target for any cancer, like epigenetic factors, modifiers, tumor microenvironment etc. Please address this issue also.

4. The data need to be validated using other sources.